# Advances in Tumor Organoids for the Evaluation of Drugs: A Bibliographic Review

**DOI:** 10.3390/pharmaceutics14122709

**Published:** 2022-12-03

**Authors:** Maritza Londoño-Berrio, Cristina Castro, Ana Cañas, Isabel Ortiz, Marlon Osorio

**Affiliations:** 1Systems Biology Research Group, Pontifical Bolivarian University (Universidad Pontificia Bolivariana), Carrera 78B No. 72a-109, Medellin 050034, Colombia; 2New Materials Research Group, School of Engineering, Pontifical Bolivarian University, Circular 1 No. 70-01, Medellin 050031, Colombia; 3Corporation for Biological Research, Medical, and Experimental Research Group, Carrera 72A # 78b-141, Medellin 050034, Colombia

**Keywords:** 3D cell culture, cancer, extracellular matrix, organoids, tumoroids

## Abstract

Tumor organoids are defined as self-organized three-dimensional assemblies of heterogeneous cell types derived from patient samples that mimic the key histopathological, genetic, and phenotypic characteristics of the original tumor. This technology is proposed as an ideal candidate for the evaluation of possible therapies against cancer, presenting advantages over other models which are currently used. However, there are no reports in the literature that relate the techniques and material development of tumor organoids or that emphasize in the physicochemical and biological properties of materials that intent to biomimicry the tumor extracellular matrix. There is also little information regarding the tools to identify the correspondence of native tumors and tumoral organoids (tumoroids). Moreover, this paper relates the advantages of organoids compared to other models for drug evaluation. A growing interest in tumoral organoids has arisen from 2009 to the present, aimed at standardizing the process of obtaining organoids, which more accurately resemble patient-derived tumor tissue. Likewise, it was found that the characteristics to consider for the development of organoids, and therapeutic responses of them, are cell morphology, physiology, the interaction between cells, the composition of the cellular matrix, and the genetic, phenotypic, and epigenetic characteristics. Currently, organoids have been used for the evaluation of drugs for brain, lung, and colon tumors, among others. In the future, tumor organoids will become closer to being considered a better model for studying cancer in clinical practice, as they can accurately mimic the characteristics of tumors, in turn ensuring that the therapeutic response aligns with the clinical response of patients.

## 1. Introduction

The International Agency for Cancer Research (IARC) reported a total of 19,429,789 new cases in 2020, of which breast, lung, colorectal, prostate, stomach, and liver cancer represent 50%. This same report reported 9,958,113 deaths, mainly from lung and colorectal cancer, which represents an incidence of this disease of 442.4 per 100,000 inhabitants [1]. In North America, the report for that year showed a total of 1,970,287 new cases, while for Latin America, 1,398,955 were reported, and in Europe, 4,042,263 cases were reported in this same period.

Cancer treatments depend on factors such as the type of cancer, its stage of progression, and the characteristics of the patient. Among the main treatments are tumor removal and conventional therapies that may or may not be combined, such as chemotherapy, radiotherapy, immunotherapy, or hormonal therapy in synergy with some alternative treatments [2]. The compounds used in cancer therapy are quite varied in chemical structure and mechanism of action, including alkylating agents, antimetabolite analogs, natural products, hormones, hormone antagonists, and a variety of agents directed to specific molecular targets. The objective of these compounds is to inhibit cell proliferation [3], generate cytotoxicity (death of tumor cells) [4], cause mechanical DNA damage [5], or be directed toward a molecular target to affect cellular function [6].

Cancer is not a disease of only one factor; at the molecular level, there is heterogeneity that can be observed from one patient to another (intertumoral) or even within the same patient (intratumoral) [1,7]. This has generated a difference in the response to conventional treatments approved to eradicate cancer, where most of the critical genes that are mutated or altered in cancer patients encode components of the pathways that regulate cellular signaling mechanisms. This variation in mutations generates heterogeneity in the therapeutic response [8]. In recent years, the understanding of cancer biology has improved, and this has led to a shift toward broader genomic tests, collaborative research, and innovative and adaptable design of clinical trials that have approached personalized medicine or precision medicine. The objective is to integrate molecular and histological analyses (analysis of mutations and signaling pathways) with clinical analyses, thus improving the ability to predict specific therapies for patients [9].

Due to the heterogeneity of tumors, the need to experimentally test the sensitivity of a tumor to anticancer compounds is reflected, as it is necessary to seek a more appropriate therapy for each individual. For this, strategies have been designed through in vitro tests where two-dimensional (2D) cultured cell lines have been the main research tool, in addition to, more recently, three-dimensional (3D) cultures, where cocultures [10]), spheroid cultures [11], organoids [12], and xenografts derived from patients are highlighted [13].

Currently, tumor organoids have aroused great interest among the scientific community, because they allow organs to be more precisely mimicked, which, in turn, will allow for better study of diseases and their possible treatments. Cancer organoids are defined as self-organized 3D assemblages of neoplastic cells, composed of a variety of cells that have the ability to grow and differentiate to mimic the histopathological, genetic, and phenotypic characteristics of the original tumor [14], and are proposed as ideal candidates for the evaluation of possible anti-cancer therapies [15]. Taking this into account, the main goal of this review is to perform a bibliometric analysis and to identify the key concepts for the design, development, and characterization of tumor organoids, as well as to determine their similarity with the tissues of origin to confirm their correspondence. Finally, this review shows the latest advances in the use of organoids for the evaluation of cancer therapies. This is a review that, to the knowledge of the authors, has not been published in the literature.

## 2. Bibliometric Analysis

The first culture protocol for long-term human organoids from neonatal or adult stem cells was performed by T. Sato et al. in 2009 [16]. From this date, research on the development of organoids as a potential tool to model the development of human diseases showed exponential growth in publications, as shown in Figure 1. The country with the greatest advances is the United States, with approximately 3500 publications, followed by China and Germany, with approximately 1000 publications, out of a total of approximately 9000 published articles. A total of 67.2% represent research articles, and 21.8% represent bibliographic reviews, highlighting studies in the areas of biochemistry, medicine, and immunology. Additionally, it was found that the number of publications that relate organoids and cancer has shown an increase since 2013, reaching 802 publications in 2021.

The effort to develop in vitro culture systems that better represent in vivo biology has identified organoids as an alternative solution. Organoids have been used for the modeling of healthy organs to understand their histology and development [17,18,19], as well as for modeling diseases, especially cancer [20,21,22,23,24] and host-pathogen interaction models [25,26,27,28,29], as well as for drug discovery through drug bioactivity and toxicity tests [30,31,32]. Of a total of 8620 publications related to organoids, as of 2021, 30.75% (2651) were related to their application in cancer, their progress and development (process to reach metastasis), or their sensitivity to drugs in a sample derived from a patient [33,34,35]. Research associated with cancer in cell lines or murine models report 28,126 and 6984 publications, respectively (“cell line” “AND” “Cancer” “AND” “TREATMENT” “for cell lines or “murine” “AND” “Cancer” “AND” “TREATMENT” for tests in mice)” while for organoids, a total of 2651 publications were reported for all years (see Figure 1b), which leads us to conclude that the topic is new in the research field.

As seen in the number of publications, the number of patents related to organoids also showed an exponential increase from 2011 to 2021 (see Figure 2), reaching a total of 219 patents published and presented for acceptance. That same year, 27 patents were accepted. Regarding patents related to tumor organoids, 18 were granted before 2021, including the construction of organoids for models of pancreatic, gastric, colorectal, ovarian, and kidney cancer, as well as methods for the generation of organoids for cancer in general (see Figure 2). This, compared to 4171 patents for cancer models (cell lines derived from various types of tumors, animal models, humanized animals, artificial neural networks among others), reflects that organoids are a novel alternative and have potential growth for modeling cancer (obtained from CANCER AND MODEL in patent inspiration).

Through the codes for the International Patent Classification (IPC codes) and the CPC codes (Cooperative Patent Classification), it was possible to identify that the main topics of these patents referred to their application in medicine (41%), active agents used in cell culture processes (11%), and support or coating of cultures and cultures in three dimensions (10%). This was reflected in the text analysis for titles and abstracts of the patents found, where the words “culture”, “tissue”, “medicine”, “cancer”, and “drugs” are related to the functions and applications of the organoids.

## 3. Organoids and Their Production

Three-dimensional cell culture is mainly represented by spheroids and organoids, the spheroids are simple structures, made up of only one type of cell, while the organoids are three-dimensional structures with complexity and heterogeneous cellular conformation, which mimic the morphology and physiology of their tissue of origin (recapitulation). The above is achieved in conditions that allow the cells that form the organoid to achieve self-renewal and provide mitogenic stimuli (culture medium that provides nutrients, growth factors that allow cell signaling, and extracellular matrix that provides support and adhesion of cells. Organoids can be initiated from embryonic stem cells, somatic adult stem cells, and induced pluripotent stem cells, although they can also originate from primary cultures of cancerous tissue samples [36]. Tumor-like organoids (also called tumoroids) are also defined as self-organized 3D assemblages of neoplastic cells derived from patient-specific tissue samples, which mimic the key histopathological, genetic, and phenotypic characteristics of the original tumor [14].

Different types of organoids have been generated from immortalized cell lines; for example, pancreatic cancer cell lines expressing the membrane marker (FC1245) [12] have been used individually or in co-culture with another line of pancreatic cancer, which express the GFP (green fluorescent protein) marker seeded in Matrigel to generate individual organoids [37]. Organoids have also been established from different types of breast cancer cell lines or primary cultures in Matrigel [38]. The above systems, with the appropriate growth factors, grow and differentiates to simulate the structure of a human tumor. In clinical research, the use of patient-derived organoids has great relevance for personalized medicine [39,40,41] For example, cells can be obtained from solid or liquid biopsies and used to generate a primary culture in a 3D matrix [14,42]. The solid biopsy has shown greater success in obtaining organoids. The tissue is digested enzymatically and/or mechanically, and is seeded on the matrix that will support the organoid. In this tissue, several types of cells that make up the tumor can be obtained, which contributes to preserving its heterogeneity [43].

Conversely, liquid biopsies have the advantage of the presence of circulating tumor cells (CTCs), which have markers and physical elements responsible for tumor spread and metastasis [44]. However, growing cultures derived from CTCs may be slow, given the low concentration of cells present in these fluids. Organoids have been generated from CTCs for pancreatic, breast, gastric, colon, and other cancers [45,46,47,48,49,50]. After the isolation of the solid or liquid biopsies, they are seeded in a biomaterial that allows biomimetics of the extracellular matrix (ECM).

## 4. Materials to Mimic the Extracellular Matrix

The ECM is a complex network of highly organized cross-linked proteins that allow cell-cell and cell-material interactions, leading to the regulation of cell proliferation, survival, differentiation, and migration [51,52,53,54]. This is why the study of their physicochemical characteristics is relevant when obtaining organoids. The biomimetics of the ECM of tumors are not simple, but are hopefully achieved from synthetic matrices. Once they are obtained, they allow the proliferation and reorganization of individual cells in a tumor in 3D [55]. Among the properties of interest of the replacement material are its chemical composition and its physical properties, such as stiffness and morphology [56].

Within the synthetic matrices are hydrogels whose networks of highly hydrophilic cross-linked polymer chains mimic the extracellular environment of body tissues [57]. Among the materials of natural origin is Matrigel^®^, which is a hydrogel of collagen and laminin derived from the secretion of Engelbreth-Holm-Swarm mouse sarcoma cells and enriched with other extracellular matrix proteins, such as type 1 collagen [58,59]. Matrigel is known as the gold standard to replace the extracellular matrix and generate 3D cultures. However, the chemical composition of Matrigel^®^ is not well defined, and [60] therefore is a source of variability in the experimental results [59]. For example, only 53% similarity has been identified between batches [61]. Furthermore, given its complexity and origin from a murine model, its use for clinical trials may be limited. This includes other materials that replace and have the same efficiency as this matrix, such as scaffolds of decellularized materials, natural biopolymers, and synthetic materials (see Table 1).

Table 1 shows natural materials derived from the decellularization of the ECM, where all the cells of a tissue or organ are eliminated while preserving their composition and morphology. This material has allowed the development of different organoids, including the liver, heart, lung, and kidney; this strategy has been explored in multiple species, including mice, rats, pigs, rhesus macaques, and humans [69]. For example, mucosal gels of the decellularized porcine small intestine allow for the formation and growth of human endoderm organoids [35,97]. Decellularized rat pancreatic tissue induces self-assembly of human pancreatic organoids [98] with an extracellular matrix of porcine brain, and it was possible to generate brain organoids from human embryonic stem cells (hESCs) [99]. Finally, the growth of different cell lines and primary cultures in scaffolds generated from the decellularized human liver was also tested [100].

The advantage of decellularization is that it can modulate cellular behavior (fixation, migration, and differentiation), has mechanical properties similar to those of its native homologs (tumor), and has important signaling molecules to modulate desired cellular behavior. However, it has limitations, such as the specificity of the components that are required for the growth of certain tissues, and the recellularization process requires a significant amount of time [97].

Natural biopolymer materials have attracted attention for imitating the extracellular matrix (biometrics). Among the compounds that have been used for their application are collagen, alginate, cellulose, hyaluronic acid, or a combination thereof. The main benefit of these natural polymers is that they are biocompatible, biodegradable, and nonimmunogenic, in addition to being able to mimic the three-dimensional structure of their tissues of origin. These materials have been attractive for developing hydrogels with the desired morphology, stiffness, and bioactivity, allowing versatility in the scaffolds that have been produced [101]. From them, organoids have been obtained for gastric, colon, breast, and testicular cancer (see Table 1), which have allowed the discovery of drugs [102], the study of cell differentiation, and the evaluation of drug delivery systems and nanoparticle-encapsulated genes [103].

Compared to synthetic polymers, strategies based on copolymers of polyethylene glycol (PEG), poly(lactic-co-glycolic acid) (PLGA) and polycaprolactone (PCL), poly(lactic-co-glycolic acid) (PLGA), polylactic acid (PLA), polyacrylamide (PAM), and several copolymers of these compounds were found [104]. This type of scaffolding has important advantages, such as its low cost and high processability (obtaining divergent structures with unique properties), by varying conditions, such as its structure, molecular weight, and incorporation of ligands, which allow the biomimetics of the physical and biochemical properties of the tumor environment. For example, PEG scaffolds combined with maleimide allow for highly reproducible in vitro growth and expansion of human intestinal cells [105]. In addition, their manufacture has low variability, which leads to high reproducibility and allows effective customization, although these materials are still far from being able to recapitulate the complexity of the matrix in vivo [104]. Additionally, a wide range of structures have been developed with varying molecular weights [90,106].

## 5. Architecture of the MEC of Tumor Cells

The MEC of solid tumors differs significantly from that of normal organs [39,107]. In general, a disorganized arrangement has been associated with tumor volume, where the ECM may represent 60% or more of this tumor mass [40,108] (Figure 2). Its composition is based on high levels of fibrillar collagens, fibronectin, elastin, and laminins in different degrees of compaction and organization [108,109,110], providing a particular environment in each type of tumor. For example, in solid tumors, the changes usually include over-expression and under-expression of molecules in the ECM [111]. In several subtypes of breast cancer, the collagen content was evaluated through Masson trichrome staining, defining three levels of high, medium, or low collagen as a function of the intensity and coverage of the staining. For the triple-negative type of breast cancer (without the presence of the three markers associated with breast cancer: estrogen receptor, progesterone receptor, and human epidermal growth factor), it presented low levels of collagen, while the subtypes of breast cancer that presented estrogen and progesterone receptors presented high levels of collagen [112]. For pancreatic cancer, an increase in collagen levels has also been reported, in addition to a greater alignment, length, and thickness in the ducts of the pancreas compared to normal ducts and benign ducts, in the context of chronic pancreatitis [113,114].

Regarding porosity, most fibrous scaffolds (such as the ECM) have pores that have a length scale similar to the diameter of a typical cell (1–10 μm) [115]. When the scaffold has pores smaller than the size of a cell, and in the absence of degradation or other forms of remodeling of the material, cell proliferation and migration can be inhibited [116]. However, when the cells are found in extracellular matrices with large pores, there are few physical restrictions, which allows more migration [117]. The strategies to modulate porosity use the concentrations of polymers such as collagen; for example, in a study by Shawn et al. (2012), in type I collagen gels, it was found that for pores between 1 and 5 μm in 1.5 mm thick gels, diameters of approximately 5 μm promoted cell migration [118]. The following figure summarizes the main differences in the architecture of the ECM of healthy tissue and of carcinogenic tissue, such as porosity and morphology characteristics of extracellular matrix proteins.

Likewise, an approach has been implemented in which collagen plugs are generated from collagen isolation, which has a pore size defined by controlling the temperature at which the collagen is polymerized [117], as is the case, for example, in the research by Infante et al. (2018), where the pore size of the scaffold was increased by reducing the temperature to 20 °C instead of 37 °C, which led to an increase of approximately two times the distance between the fibrils and, thus, the increase in pore size [119]. In addition, in order to promote pore formation, different porogenic agents have been used, such as poly(ethylene glycol) or dimethyl ether, which can generate pores of approximately 100 µm. These types of porous materials have been used to simulate the microenvironment of pancreatic tumors [120].

The architecture of the ECM influences the biological functions that lead to tissue development [121]. Thus, apart from pore size, physical properties such as stiffness and chemical properties (composition and ligands) are related to the ability of tumor cells to promote tumor growth.

The quantification through atomic force microscopy of the elastic modulus of Young allows for the determination of the stiffness of the tissues, which varies significantly between them and is inherently related to the function [122]. From this technique, it has been determined that the ECM of tumor tissues is stiffer compared to normal tissues [123]. In breast cancer, for example, it has been found that the most aggressive cancer subtypes have greater matrix stiffness (>5 kPa), while for normal tissue, this stiffness is ~0.2 kPa [124]. This stiffness has been associated with the incorrect folding of fibronectin in the tumor during the development of breast cancer, or with the increase in the number of collagen fibers [125]. For glioma cells, matrices with a stiffness that varied from an order of magnitude below normal brain tissue (0.08 kPa) to three orders of magnitude above normal brain tissue (119 kPa) were evaluated, which led to the conclusion that the increase in the stiffness of the ECM can induce greater cell propagation, faster motility, and greater proliferation [126]. On the other hand, tissue stiffness can be related to treatment efficiency, as reported in the evaluation of photothermal therapy with multiple-walled carbon nanotubes for a mouse xenograft with epidermal carcinoma [127]. It can also be related to changes that would be generated by structural variations and thermal damage, as well as the denaturation of proteins promoted by the treatment. This same approach could be implemented in organoids in vitro, evaluating the integrity of the membrane (in its mechanical properties such as stiffness) at different treatment times.

Regarding the chemical properties, the composition of the ECM could generate cytotoxicity or immune reactions. Thus, natural scaffolds have an advantage due to their low cytotoxicity and ability to mimic the interactions between cells (due to their ligands) and the prevalence of growth factors, but within their limitations is the possibility of modifying these systems to incorporate additional functional groups [128]. On the other hand, synthetic scaffolds can be easily manipulated and can mimic the chemical characteristics of the ECM. For example, scaffolds can incorporate ligands combining biologically active peptides or proteins to contribute to cell signaling [116], but their biomimicry is more limited. Scaffolds derived from a natural acellular matrix can retain the unique biomechanical property, composition, and structure of MECs [129], imitating the precise architecture of the tumor tissue, but this is a time-consuming process [97].

Another factor to have in mind is the tumor microenvironment. According to Figure 3, the tumor microenvironment is composed of multiple supporting cells, including cancer-associated fibroblasts, non-tumor cells, and immune cells such as lymphocytes, neutrophils, dendritic cells, and monocytes. The presence or absence of these cells indicates the stages of the disease. In the initial stages, immune cells contribute to the elimination of cancer cells by recruiting cytotoxic cells, such as natural killer (NK) cells and CD8+ T cells [130]. When the tumor is advanced, the presence of macrophages is associated with a poor prognosis [131].

Cancer-associated fibroblasts support tumor growth and metastasis, and have been associated with drug resistance [132]. Given their importance, some studies have incorporated this type of cell in order to more accurately mimic the tumor microenvironment. For example, the research by Liu et al. (2021) found that establishing three-dimensional (3D) co-cultures of organoids derived from primary liver tumors with cancer-associated fibroblasts decreased response to treatment with Sorafenib, regorafenib, or 5 -fluorouracil [133]. Likewise, the presence of cancer-associated fibroblasts promotes the tumor progression in lung cancer organoids, as they induce a better microenvironment [134].

## 6. Techniques for the Development of Tumor Organoids

The initial conditions of the organoid contribute to its variability, including the initial cell population, its positioning and aggregation [135], its niche or extracellular environment, its physicochemical characteristics, and its culture conditions. At first, organoid cultures may start as spheroids or agglomerated cells. After cell differentiation, promoted by growth factors, they acquire their organoid characteristics [136]. These, once spread, are maintained under controlled CO_2_ and temperature conditions (5% *v*/*v* of CO_2_ and 37 °C). In systems used for organoids, they are submerged in culture medium and microfluidics, and cultured in an air–liquid interface, or in bioreactors [13].

### 6.1. Submerged Culture in Scaffolding

The first method consists of the immersion of the organoid promoter cells or spheroids of tumorigenic cells, supported by a scaffold that will allow their growth in 3D. These are immersed in the culture medium, then supplemented with growth factors that can vary according to the type of tissue. The culture media that have been reported for the cultivation of organoids are Dulbecco’s modified Eagle medium (DMEM), Eagle’s minimum essential medium (EMEM), Roswell Park Memorial Institute (RPMI) medium, and Ham’s F12 culture medium (F12). This is conducted in the presence or absence of fetal bovine or equine serum and antibiotics, in addition to other supplements, such as L-glutamine, HEPES buffer, and GlutaMAX supplement; however, the critical components of the organoid media are a set of growth factors that include epidermal growth factor (EGF), fibroblast growth Factor 10 (FGF10), hepatocyte growth factor (HGF), R-spondin 1, and noggin. Figure 4a shows the general scheme of this technique for the development of organoids.

For example, in the development of hepatic organoids, Lugli et al. (2016) determined that the growth of these organoids was stimulated by R-spondin 1 and noggin, while in their absence, they partially differentiated into hepatocytes. For pulmonary organoids, changes in morphology were found depending on the presence of EGF, where the organoids were significantly smaller in the absence of this growth factor [137,138,139]. However, it is necessary to improve the understanding of a possible optimal cocktail of growth factors, depending on each tissue, to be closer to the sample of interest [140].

### 6.2. 3D Microfluidics

Three-dimensional microfluidic cultures consist of organoids generated from cells embedded within a collagen gel in the middle of a microfluidic culture device. They consist of different materials (polydimethylsiloxane, silicon, glass, polycarbonate, polymethylmethacrylate, polystyrene, cyclic olefinic polymers, and polyimide) on which straight channels or more complex structures are molded. Through these, the fluid that will pass over the microchannels is pumped [141]; in this case, supplemented culture medium (as presented in the submerged 3D cultures) flows from the channels located on both sides of the central region [142], as shown in Figure 4b. This system is particularly beneficial for its application in cancer, given the interactions that occur between the tumor microenvironment and the tumor and the possibility of regulating them [143].

Applying this strategy, Wang et al. (2013) cultured the A549 tumor cell line submerged in BME (R&D Systems, MN) as a substitute for ECM to achieve three-dimensional growth in the cell culture chamber, which was connected to syringe pumps through each of the inlets to drive the fluid flow at a rate of 0.1 µL/min of medium or drug. The experimental results showed that this is a good model for 3D growth, allowing the evaluation of protein expression [144]. Likewise, for breast cancer, organoids were generated on a chip by coculturing phenotypically normal and diseased cells and tumor nodules, finding that in this system, it was possible to recapitulate the luminal environment of the breast [145].

### 6.3. Air–Liquid Interface Culture

Organoid culture in an air–liquid interface (ALI) is a method that consists of seeding the set of cells derived from the tumor in a transwell dish. These cells are exposed to the culture medium at the base of the dish, which can acquire oxygen through a matrix that surrounds them and interacts with the air [146] (see Figure 4c). From this platform, it has been possible to successfully obtain organoids for colon and pancreatic cancer, among others [147,148]. For example, the normal and tumor tissues of the patients were included in a collagen gel and cultured using an ALI culture system. Additionally, renal cell carcinoma was cultured on this platform in the presence of cells of the immunological system by Esser et al. (2020), where the tissue was fragmented and cultured in the collagen-based ALI system, and organoids were generated for which it was possible to determine the correspondence with the tumor of origin by IHC staining, RNA sequencing, and drug response [149]. The main advantage of this methodology is that not only the genetic alterations of the tumor, but also the complex cellular composition and architecture of the tumor environment, can be recapitulated.

### 6.4. Bioreactors

Bioreactors are generally defined as devices in which biological and/or biochemical processes are developed under close supervision and strictly controlled environmental and operational conditions [150]. These restrictions allow the possibility of controlling environmental conditions, such as oxygen stress, pH, temperature, shear stress, sterility [151], aeration, and nutrient distribution, which, in turn, allow the growth of complex structures. However, they must be designed based on a comprehensive understanding of the biological and engineering aspects; that is, the operating conditions must be specified (see Figure 4d) (119). To counteract the low efficiencies in seeding and nonuniform cell distributions within the scaffolds, bioreactors are presented as an alternative, given the possibility of controlling culture conditions; with them, organoids have been generated through bioreactors.

For example, Skardal et al. (2014) developed a liver tumor organoid system in which HepG2 cells and HCT-116 metastatic colon carcinoma cells were cultured in rotating bioreactors with hyaluronic acid and gelatin microcarrier beads, which led to the initiation and growth of cell aggregates [152]. Lancaster et al. (2014) developed compared the static culture of colorectal cancer samples with a perfusion bioreactor, showing that the organoids obtained by the bioreactor maintain the architecture of the tumor tissue and the densities of the proliferating tumor cells to a significantly higher degree. In addition, static cultures emulate the characteristics of the ECM, which can contribute to the evaluation of the response to the tumor drug in a specific context of the patient [153].

### 6.5. In Silico Models

Tumor organoids also allow the modeling of the morphologies of experimental multicellular culture systems, serving as a basis to preselect possible experiments before performing them (129) and providing optimization for preclinical trials. In principle, they simulate cell growth and morphology of the scaffolds that would support the organoid; for example, Pang et al. (2019) designed a micro-scaffold with computer-aided design (CAD) software to find the optimal characteristics of the matrix that would support the organoid, and in this cell line, Hep G2, TMNK-1, and Swiss 3T3 were cocultured. This design was shown to be more efficient in achieving a high number of retained cells and liver functions [154]. Likewise, a prediction of the organoid model of intestinal tissue determined that these organoids cannot grow in rigid and flat substrates, and that the expansion of stem cells in an organoid depends significantly on its biomechanics [155].

Organoid culture models are being actively developed to improve the pharmacogenomic similarities between preclinical models and tumors; for example, Kong et al. (2020) integrated pharmacological data derived from in vitro tests for colorectal and bladder cancer tumor organoids. Through network-based methods and machine learning, it predicted patient responses to medications based on the interaction between the main protein networks and drugs [156]. Kather et al. (2018) designed an *in silico* model, computationally based on human colorectal cancer, that also includes lymphocytes, macrophages, fibrotic stroma, and necrosis, and can develop large tumors of more than 10^6^ cells in a few minutes with standard computer hardware. This model accurately recapitulates the behavior at the cellular and tissue levels based on changes in the structure of the cell or the extracellular matrix [157].

## 7. Techniques to Evaluate the Correspondence between Organoids and Their Tissue of Origin

In general, organoids are generated from various patient samples obtained through a variety of collection methods, including surgical samples, core biopsies, and samples of malignant fluids. It is often reported that the success rate of organoid generation is greater than 70% [158]. The extrapolation of the results of the model systems to humans has become a major obstacle in the process of drug discovery [159], since it is necessary to mimic the histopathology, genetics, phenotype, and pharmacodynamics of the tumor of origin, as well as to maintain marker expression (immunofluorescence) [97]. Therefore, once the organoids are obtained and established, it is necessary to carefully evaluate their correspondence with their system of origin through different techniques, as shown in Figure 5 and detailed below.

### 7.1. Histological Techniques

Comparative histology can be performed after fixing organoids and original tissue samples, paraffin embedding, and sectioning, followed by immunohistochemistry [160]. The histological analysis allows microscopy and specific stains to identify different types of cells and tissues, in order to obtain important information on characteristics such as cell shape and structure. Likewise, immunohistochemical analysis allows the determination of the presence and specific level of cellular proteins, such as p53, human epidermal growth factor receptor 2 (HER2), epidermal growth factor receptor (EGFR), proliferating cell nuclear antigen (PCNA), among others. Itprovides an idea of the quantification of these proteins at the cellular level [48,158,161].

Regarding the histological characteristics of organoids and cancerous tissues, it should be considered that the U.S. National Cancer Institute classifies cancer by the type of tissue where it originates (histological type) and by the site where it was first developed [162]. Thus, carcinoma is a malignant neoplasm of epithelial origin or cancer of the internal or external lining of the body, and is classified as adenocarcinoma if it develops in an organ or gland and squamous cell carcinoma if it originates in the squamous epithelium. Sarcoma refers to cancer that originates in the supporting and connective tissues, while myeloma originates in plasma cells of the bone marrow. Leukemia and lymphoma develop in the glands or nodes of the lymphatic system [163].

There is another classification in which the degree of cancer is defined, known as histological grade; this is related to cell morphology and provides prognostic information on the medical status of the patient [164].

Additionally, histological techniques allow researchers to compare, for example, breast cancer [165], colorectal [166], gynecological [167], hepatic organoids [168], and others. Organoids were compared with the native tissue through staining by bright-field microscopy and staining with hematoxylin–eosin, reflecting that the phenotype of the organoids often coincided with its original counterpart. In the research of Michalopoulos et al. (2001), in addition to histological characterization, immunohistochemical analysis was performed, in which biomarkers such as the estrogen receptor, P53, progesterone receptor, and HER2 were determined, finding that their profile was similar to that of the original samples. It was concluded, however, that histological analysis may fail to accurately categorize well-differentiated organoids. In addition, other experiments have evaluated the ultrastructural characteristics of the tissue by transmission electron microscopy [168].

### 7.2. Cytological Techniques

Flow cytometry is a technique that allows the identification and classification of the types of cells present in tissue through their classification by membrane markers or cell characteristics of present populations (size, cytoplasm constitution, amount of DNA) [12,169]. Through these tests, the identification of cell populations present in organoids has been achieved to distinguish between normal cells and cancer cells, as well as to determine the correspondence in terms of organization and expression of certain proteins of interest of the organoids and their tissue of origin [115].

Flow cytometry analysis has determined the heterogeneity of tumor cells through their classification by membrane markers [170], and has confirmed the presence of progenitor cells. For example, in liver cancer organoidsthe cells expressed stem cell surface markers (CD24 and CD44) [171,172,173]. This is important because stromal cells in the tumor microenvironment are crucial for the development and progression of cancer. For example, Kim et al. (2019) evaluated the tumor microenvironment of their organoids and found that their cells contained heterogeneous characteristics in terms of the expression of membrane markers; some were a mixture of p63 + cells and p63− cells, others were only CK7+, other organoids were only CK5/6+, and others comprised a mixture of CK7+ and CK5/6+ cells, with different spatial patterns of cell types [174].

### 7.3. Molecular Techniques

Cancer genetics are also important for the generation of organoids, genomic mutations in tumor suppressor genes, proto-oncogenes, genes involved in chromatin remodeling, mutations in genes involved in the cell cycle or signaling pathways, and mutations in genes associated with metabolism [175]. Differential expression or modification has also been reported in the expression of genes related to metabolism or control of cell growth; epigenetic modifications, such as methylation, acetylation, or other modifications of chromatin; alterations or posttranslational regulation given by mRNAs; and structural or numerical variations, such as changes in chromosome stability, chromosome damage, and changes in the number of chromosomes [8].

Thus, these fundamentals of cancer genetics contribute to the evaluation of these expression levels through the omics sciences (genomics, transcriptomics, proteomics, epigenomics). Genomics, on the other hand, allows the comprehensive detection of genomic alterations in human somatic cells, including point mutations, chromosomal rearrangements, and structural variations, through next-generation sequencing and complete exome sequencing [74], techniques that have been used for the evaluation of organoids to determine their correspondence with tissues of origin [49,158,176]. In turn, the evaluation of RNA expression profiles has been carried out through RNA-seq transcriptomic analysis or through the sequencing of a single cell, which has allowed the comparison between the organoids and their tissue of origin [48,161,176]. Proteomic profiles have also been evaluated by obtaining total proteins and their subsequent identification to identify changes in quantitative protein profiles and posttranslational modifications [176].

A multi-omic analysis (evaluation of the genome, proteome, epigenome, and metabolome) of intestinal organoids accurately recapitulates epithelial homeostasis in vivo, highlighting that although each technique is informative in itself, by performing multiple complementary techniques, they will strengthen each other [176]. Likewise, the transcriptomic analysis of individual cells highlights the intratumoral variation; for example, Hanbing Song et al. (2022) reported heterogeneous cell states in prostate epithelial cells that corresponded to the tissue of origin [177].

To corroborate these correspondences, a significant number of studies have been generated which evaluate different variables of the tissue of origin and the cells established in the organoid. For example, Pasch et al. (2019) generated cancer organoids derived from samples of colorectal adenocarcinomas, pancreatic and lung tumors, neuroendocrine tumors of various organs, and other tumors. In addition, through molecular tests, they corroborated that the generated organoids represent the cancers that were derived [158]. Figure 3 summarizes the collection, establishment, and techniques used to evaluate tumor organisms.

## 8. Organoids as a New Strategy for Drug Evaluation

Given the complexity and low response of patients to cancer treatments, in vitro and in vivo tests have represented an important aid at the time of diagnosis and choice of the most relevant therapies, in addition to ensuring that the molecules have the pharmacology and adequate activity of the biological system [178]. Thus, the identification of the preclinical model should reflect the in vivo conditions of patients to ensure that the results provide sufficient evidence to advance toward clinical development; consequently, different models have appeared, with their advantages and disadvantages. The following figure (Figure 6) shows the advantages and disadvantages of each technique.

### 8.1. 2D Culture

The use of two-dimensional (2D) cultures consists of obtaining cells in adherent conditions in which the cells adhere to a glass or plastic dish or in a suspension, and have been used for cancer research and drug discovery of new therapies, given their ease and low cost. These tissues do not represent the structure in vivo, because the cell–cell and cell relationships with the environment do not correspond to normal tissue conditions [179]. In addition, cells cultured in two dimensions lose their polarity, changing the cellular response to different stimuli [180]. The above can simplify that 2D cultures of cancer cells are different from human tumor cells.

### 8.2. Animal Models

Cancer research animals’ models have been used for cancer drug discovery testing in mainly mice and rats; nevertheless, there are other systems, such as hamsters, rabbits, mini pigs, dogs, sheep, goats, horses, primates, and zebrafish [181]. These models incorporate the organization of the tissue in 3D and offers analysis at the system level; however, limitations are the high cost and/or the use of immunocompromised animals. The interaction between the immune system and cancer processes is highly important because, as animals did not spontaneously express cancer, immune cells can act against tumors by absorbing, presenting tumor antigens, releasing cytokines, or directly killing tumor cells during the process of creating cancer models [182].

In addition, the use of these models goes against the new trend that aims to reduce or eliminate the use of animals in preclinical tests. For instance, the average rate of successful translation from animal models to clinical cancer trials is less than 8%. Animal models are limited in their ability to mimic the extremely complex process of human carcinogenesis, physiology, and progression. Therefore, the safety and efficacy identified in animal studies are generally not translated to human trials [183]. In other words, animals are not accurate enough to model human cancer.

To overcome the disadvantages of producing cancer in healthy animals, researchers have worked with the use of xenograft models, where developed cancer tumors or cancer cells are grafted into animals [184]. Nevertheless, these models bring their own limitations, as they require the use of immunodeficient hosts [185], and the animals require specialized facilities and personnel for the proper course of the investigation.

Recently, researchers such as Sugimoto et al. (2022) have compared mouse models and intestinal epithelial organoids in in vitro cultures, where animal studies presented high mouse maintenance costs, poor human–mouse correspondence, and variations in the molecular regulation of cellular processes [186].

### 8.3. 3D Models

Within the three-dimensional cultures are spheroids and organoids, each providing the opportunity to mimic a fluid cell–cell interaction and the possibility of biomimicry of the extracellular matrix (please see Figure 7), which allows a more precise approach to the structures in vivo. In addition, they allow the generation of cocultures and demonstrate the architectural complexity of the development of human tissues and organs; however, the difficulty in standardization continues to be a limitation for the use of these cultures as clinical tests, and many of these are still under investigation [135].

Given that organoids recapitulate the biological and molecular characteristics of the tumor of origin, as explained in the previous sections, they have the potential to be used for the generation of preclinical models aimed at improving treatments or as predictors of the response to individualized treatment. The evaluation of drugs or therapies in organoids has shown that they are more sensitive to 2D cultures by presenting a structure that more closely resembles the cellular structure in vivo, including cell–cell interactions and cellular diversity [187]. Moreover, patient-derived organoids are translational and more accurate than animal models. Table 2 compares the use of organoids and 2D cultures in the evaluation of anticancer compounds.

## 9. Evaluation of Anticancer Therapies in Organoids

Organoids can be postulated as possible platforms for the evaluation of drugs specifically for patients (personalized medicine). Once the organoids were established from autologous cells of patients, a correlation was achieved between the response to the treatments at the in vivo level and the organoids, correlating the efficiency of the treatments or possible drug resistance. The response of organoids to drugs can be evaluated through numerous cell viability or proliferation assays based on cellular functions [12,81], which allow the determination of the inhibitory concentration (IC50). This is obtained once the organoids are exposed to a wide range of concentrations of the drugs to be tested, and the subsequent determination of the number of viable cells and/or organoids has occurred. In this way, lethal and sublethal concentrations can be determined, and variations in these concentrations can determine the resistance or sensitivity of the organoids to the evaluated drugs [12].

The determination of cellular metabolism can be evaluated through the quantification of ATP present as an indicator of metabolically active cells (CellTiter-Glo-Luminescent Cell Viability Assay). Thus, organoids derived from castration-resistant prostate cancer showed sensitivity to enzalutamide with a low IC50 (50 nM). This was corroborated by the xenografts derived from these same organoids, which were highly sensitive to enzalutamide. Recapitulating the in vitro results [48], mitochondrial activity can also be evaluated through the MTT colorimetric assay to determine the lethal concentrations of the drugs tested, and has been one of the most widely used techniques for this purpose [195]. Histological analyses have also allowed the evaluation of the effect of drugs on organoids, evaluating post-treatment cell morphology and measuring the diameter of exposed organoids [161], where the decrease in diameter is correlated with the success of the tested therapy.

In addition, the expression of genes associated with the efficiency of therapy or treatment has been evaluated; for example, organoids derived from renal cancer biopsies were exposed to conventional directed therapy (Sunitinib and Tensirolimus, SU11274, Foretinib, Cabozantinib, and Leucinib in combination with Everolimus). After the exposure of healthy cells to the organoids, the activity of Caspase-3 (as an indicator of apoptosis) and genes such as pAKT S437 and pERK T202/Y204 was evaluated through Western blotting, since the expression of these is related to the efficacy of the therapy. The results showed that there was no therapeutic effect on healthy cells, but drugs such as Foretinib and SU11274 generated changes in the expression of these genes, which could indicate that these organoids represent a new approach for therapy decision-making [22]. Table 3 shows recent advances in the evaluation of anticancer therapies using organoids.

## 10. Conclusions and Perspectives

The search for new alternatives for cancer treatment has contributed to the investigation into cellular models that reduce the risks associated with experimental treatments to combat cancer. This is how organoids emerged as novel platforms for these purposes. According to what was found in this review, these have the potential to replace 2D systems and murine models. Their low cost is a major advantage, as is their recapitulation of the tissue of origin, by which organoids can perform biomimetics of living tissues, presenting a gene-based and phenotypic histological correspondence with the tumor of interest.

The challenges for the application of organoids are the understanding of the scaffold that supports the tumor cells and the factors that influence their growth and differentiation, such as cell–cell interaction, the interaction of cells with the ECM, and the different growth factors required. Modeling these aspects is important for developing correspondence with the systems that the organoids represent. In the future, the use of animal-derived matrices (such as Matrigel, among others) to create organoids should be avoided, because it generates variability (making the system difficult to characterize). There is also the need to find natural or synthetic alternatives to the ECM that lead to proper growth and signaling for cell growth and differentiation, as an alternative for the actual strategies, as can be observed from the rapid growth of papers and patents on this topic in the last 20 years. Between them, different materials, such as scaffolding, have been sought. This has encouraged the development of other polymeric materials, such as those discussed in Table 1. Regarding the origin of cells and cell types, organoids derived from patient tissue are more accurate in the biomimetics of tumor tissue, decreasing the need to use animals in cancer therapy evaluations. Animals fail to fully model the physicochemical and biological features of human cancer.

Organoid technology is a great advance in the field of basic and clinical research, and it has allowed us to answer detailed biological questions, predict patient outcomes, and identify the most effective pharmacological compounds, contributing to clinical decision-making during the course of therapy. In the medium term, organoids will be used in clinical practice as representations of organs, tumors, or as disease models, since they are a more precise alternative to conventional models given the recapitulation of biological conditions. In the long term, the information provided by organoids will be of great help for the creation of in silico models that allow the rapid discovery of new therapies, in this case, without the use of animals, scaffolds, or human cancer cells.

## Figures and Tables

**Figure 1 pharmaceutics-14-02709-f001:**
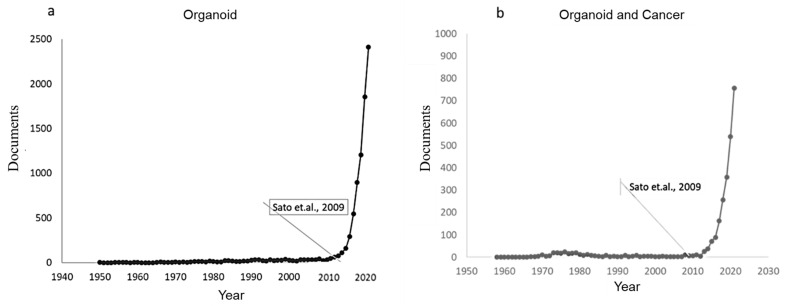
Documents published with reference to the topics of interest. (**a**): Publications over time of the organoid topic. (**b**): Publications over time of the organoid and organoid/cancer topics. For this, a search was performed in the SCOPUS, Nature, and Science-direct databases using the keywords “Organoid” AND “Cancer” AND “Characterization” as the search equation, and selecting the documents of interest from the titles and abstracts [16].

**Figure 2 pharmaceutics-14-02709-f002:**
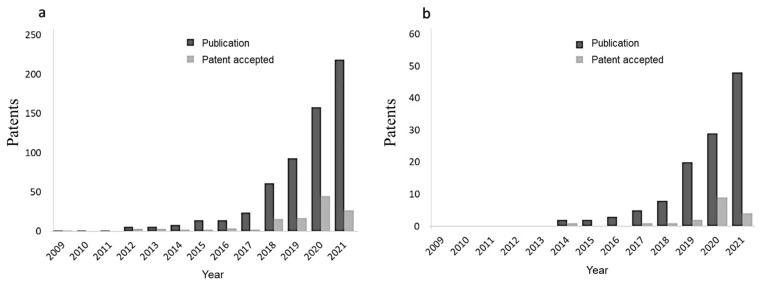
Number of patents applied for and published per year. (**a**): Patents corresponding to organoids. (**b**): Patents corresponding to organoids for cancer. The data were obtained through “Patent inspiration”, adding “Organoid”, “AND”, and “Cancer” as filters.

**Figure 3 pharmaceutics-14-02709-f003:**
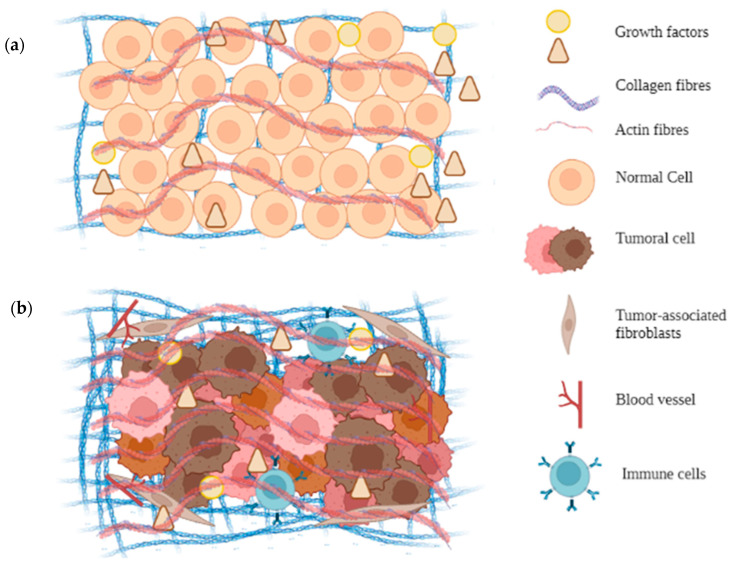
Comparison between the architecture of healthy cell tissue and tumor tissue. (**a**). Architecture of a healthy tissue and (**b**). architecture of a solid cancer tumor. Created with BioRender.com.

**Figure 4 pharmaceutics-14-02709-f004:**
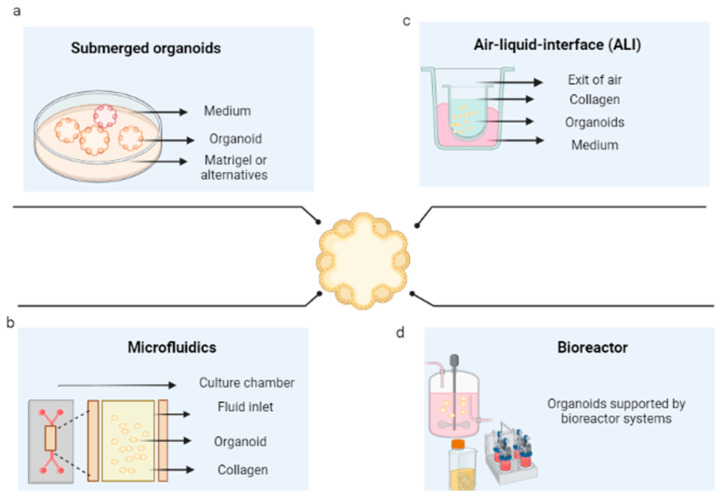
Organoid culture techniques. (**a**) Culture of organoids immersed in an extracellular matrix. (**b**) Organoids cultured in microfluidics. (**c**) Organoids cultured in an air–liquid interface (**d**) Organoid culture in bioreactors. Created with BioRender.com.

**Figure 5 pharmaceutics-14-02709-f005:**
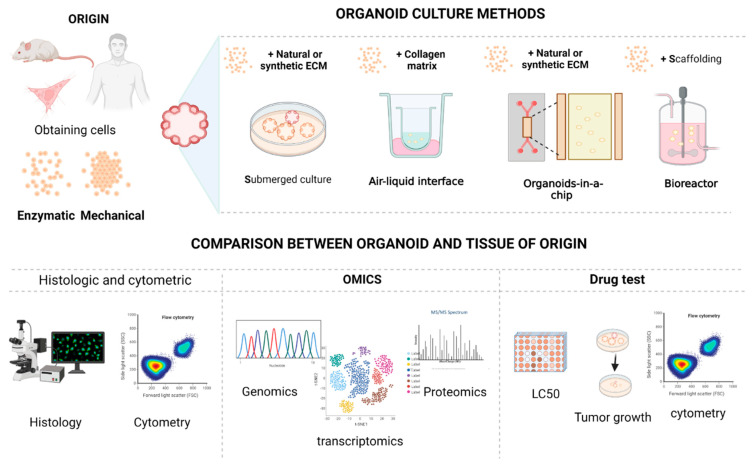
Process for obtaining and seeding tumor organoids and comparison techniques of organoids with their provenance system. Created with BioRender.com.

**Figure 6 pharmaceutics-14-02709-f006:**
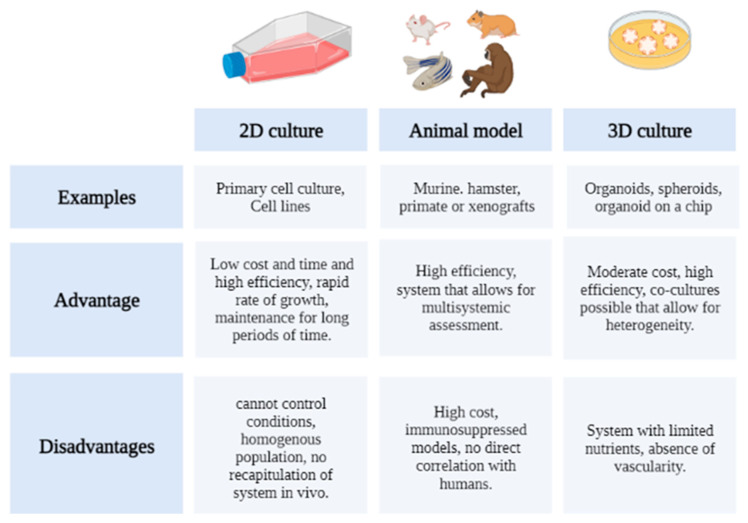
Advantages and disadvantages of commonly used systems for the evaluation of anticancer drugs. Created with BioRender.com.

**Figure 7 pharmaceutics-14-02709-f007:**
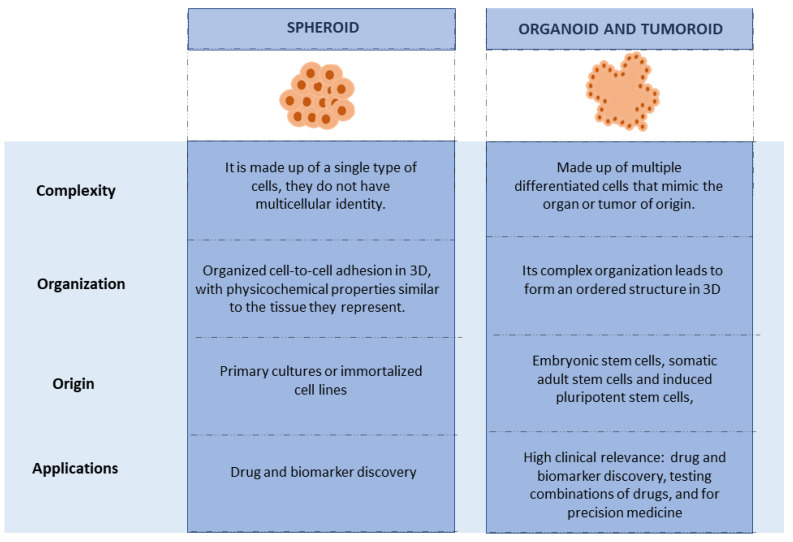
Comparison between organoids and spheroids.

**Table 1 pharmaceutics-14-02709-t001:** ECM biomimetic materials for the development of tumoroids.

Scaffolding	Origin	Composition	Characteristics of the Material and Function in the MEC	Technological State	Cancer Organoid Generated
Decellularized matrices	Natural	Engelbreth-Holm-Swarm mouse tumor (EHS matrix)	The basement membrane is structurally a thin layer of ECM composed by collagen type IV, entactin, perlecan, and laminin-like extracellular environment of the basement membrane [62].	Commercial: Matrigel ^®^ Research: Pinho et al., (2021), Lee et al., (2020), Saito et al. (2020) [63,64,65]	Cancers: colorectal, uterus, bile duct tumors, prostate cancer, lung, etc.
	Natural	Decellularized extracellular matrix from healthy liver and tumor liver were obtained by detergent–enzymatic treatment	Decellularizing methods vary by the origin (tissue) and the specific physical, chemical, and enzymatic methods used. The acellular has functions in homeostasis, regeneration, tissue, and organ development through cell surface molecule interaction and storage of growth factors [66].	Research D’Angelo et al. (2020) [67]	Liver cancer
		Decellularized extracellular matrix were derived from both peritoneal metastases and normal peritoneum		Research: Varinelli et al. (2022) [68]	Peritoneal cancer
		Mucosa/submucosa of the decellularized porcine small intestine.		Research: Garreta et al. (2017) [69]	Endoderm-derived organoids, such as gastric, hepatic, and pancreatic
		Decellularized colorectal cancer matrices		Research Marques- Magalhães et al. (2022) [70]	Colorectal cancer
		Porcine lung decellularized extracellular matrix		Research: Park et al. (2021) [71]	Lung cancer
		Decellularizing and delipidating a porcine breast tissue whit addition of gelatin methacrylamide and alginate		ResearchBlanco-Fernandez, et al. (2022) [72]	Breast Cancer
Polymeric materials	Natural	Rat tail collagen (Type I collagen)	Collagen is a fibrous protein that consists of three α-chains that can combine to form a rope-like triple helix. The collagen fibers both strengthen and help organize the matrix, and rubber-like elastin fibers give it resilience [73].	Commercial: Corning™ BioCoat™ Collagen I Research: Mollica et al., (2019), Mitaka et al., (2002) [74,75].	Liver and breast cancer
		Hyaluronic acid (HA)	HA is a linear polysaccharid, found in various tumor tissues, and is considered a tumor-associated extracellular matrix [76].	Research: Narkhede A, et al. (2020) [77]	Metastatic breast cancer
		Hyaluronan and matrix metalloproteinase-cleavable	Linear polysaccharide with disaccharide repeats of d-glucuronic acid and N-acetyl-d-glucosamine abundantly expressed in ECM. This organizes and maintains the structural integrity of extracellular and pericellular matrices [78].	Research: Baker et al. (2022) [79]	Breast cancer
		Alginate	Anionic polysaccharide, composed of α-l-guluronic acid and β-d-mannuronic acid, is an immobilization matrix for cells, but is unable to specifically interact with mammalian cells [80].	Research: Fang et al. (2021) [81]	Breast cancer
				Research: Qiu et al. (2021) [82]	Liver cancer
		Cellulose	Cellulose is a linear polymer of glucose, which serve as potent tissue scaffold; it is biocompatible and stable over time [83].	Research: Curvello et al. (2021) [84]	Pancreatic cancer
		Chitosan	Chitosan of poly (1, 4 D-glucosamine), a partially deacetylated derivative of chitin, is similar to glycosaminoglycan (GAG), one major component of ECM in the cancer environment [85].	Research: Han et al. (2016) [86]	Lung cancer
		Gelatin	Gelatin is a molecular derivative of collagen obtained through the irreversible denaturation of collagen proteins [85].	Research: Luo et al. (2020) [87] Shengyong et al., (2019) [88]	Colorectal cancer
	Synthetic	Polyethylene glycol (PEG)	PEG facilitates the biocompatiblility of3D culture. Moreover, PEG has modular bioactive features like cell-mediated material degradation [89].	Research: Gjorevski et al., (2017) [90]	Instinct Tumor
				Research: Xiao et al., (2018) [91]	Glioblastoma
		Poly Lactic-co-Glycolic Acid (PLGA) and polycaprolactone (PCL), Poly (lactide-co-glycolide) (PLG)	PLGA is a copolymer of poly lactic acid (PLA) and poly glycolic acid (PGA). PLGA is biocompatible and biodegradable, exhibits a wide range of erosion times, has tunable mechanical properties, and, most importantly, is an FDA-approved polymer [92].	Research: Rijal et al. (2017) [92] Research: Dye et al. (2020) [93]	Breast cancer Lung cancer
		Poly (ε-caprolactone) (PCL)	PCL is a biodegradable polyester. As a biocompatible, biodegradable, and bioresorbable polymer, PCL has also found plenty of uses in the form of microparticles and scaffolds [94].	Research: Sims et al. (2014) [95]	Breast cancer
		Polylactide (PLA)	PLA is a versatile commercial biodegradable thermoplastic based on lactic acid. PLA has been an extensively used as a biomaterial applied in medicine, as it is a biodegradable and biocompatible synthetic polymer.	Thankuri et al., [96]	Various

**Table 2 pharmaceutics-14-02709-t002:** Comparison between 2D cultures and organoids for the evaluation of anticancer drugs.

Researchers	Objective	Model	Findings	References
Casey et al. (2016)	Chemotherapeutic evaluation with doxorubicin	HeLa cell line	Lower toxic response of cells cultured in 3D, reduced drug efficiency in 3D.	[188]
Jiguet et al. (2014)	Chemotherapeutic evaluation with cisplatin and radiotherapy	Glioblastoma. organotypic culture systems	In 3D morphology, organization and markers better recapitulate the original tumor. 3D cultures are more resistant to the evaluated therapies.	[189]
Kim et al. (2020)	Chemotherapeutic evaluation with staurosporine, irinotecan, 5-fluorouracil or SN-38	Cells derived from patients cultured in 2D, spheroids, and organoids	It was determined that 3D models are advantageous over traditional 2D cultures for the detection of pharmacological compounds.	[190]
Cannon et al. (2017)	Chemotherapeutic evaluation with treatment with XL147	2D models and organoids of breast cancer	3D models offer a superior method to analyze the efficacy and resistance to drugs within in vitro studies.	[191]
Chae et al. (2020)	Biomimetic characteristics of the two crops.	2D models and organoids of kidney cancer derived from patient cells.	Compared with 2D culture, 3D organoid cultures preserved the characteristic lipid-rich clear cell morphology of renal cell carcinoma.	[64]
Elbadawy et al. (2019)	Response to combined treatment with cisplatin, vinblastine, gemcitabine, or piroxicam	Urothelial carcinoma in 2D dog culture compared with organoids derived from dog bladder cancer	It was found that 3D organoid cultures were more suitable for identifying novel biomarkers in bladder cancer with human muscle invasion.	[192]
Duarte et al. (2018)	Comparison of the number of DNA copies in organoids	Organoids of breast tumors and 2D cell lines with patient tumors	Greater agreement between tumor samples and organoids on 2D cell lines indicates that organoids may be a more suitable culture system for modeling patients.	[193]
Vincent-Chong et al. 2020	X-ray	2D model, 3D model, and xenografts of oral squamous cell carcinoma	Radiation had no effect on immunocompromised mice with xenografts, but affected 2D and 3D cultures.	[185]
Guillen et al. 2022	Eribulin	Organoids and xenografts of breast cancer	Correspondence was found in the evaluation of drugs in organoids and xenografts.	[194]

**Table 3 pharmaceutics-14-02709-t003:** Evaluation of anticancer drugs in different types of patient-derived organoids (PDO).

Organoid	Medication	Techniques	Findings	Ref.
PDO head and neck cancer	Cisplatin, docetaxel	Clonogenic cell survival and determination of half-maximal inhibitory concentrations (IC50)	Response of pharmacological treatment depending on the dose and type of medication. The findings were translational to clinical studies.	Tanaka et al. (2018) [196]
PDO lung cancer	Inhibitors of the human epidermal growth factor receptor 2 (HER2)	High-throughput screening to evaluate organoid growth inhibition and cell viability through ATP quantification	The organoids were suitable for evaluating targeted molecular drugs that simulate the pathological conditions of the patient.	Takahashi et al. (2019) [197]
PDO ovarian cancer	Combined carboplatin/paclitaxel therapy	Cell-Titer Glo2.0 Assay	Heterogeneity was found in the response to the drug between patients and intrapatients. Results were similar to those found in ovarian cancer patients, and the organoids are useful for directed therapy.	Witte et al. (2020) [198]
PDO gastric cancer	Cisplatin, oxaliplatin, and irinotecan		Standard dose-response curves to current cytotoxic therapies for gastric cancer.	Gao et al. (2018) [199]
PDO liver cancer	Screening of 129 anticancer compounds approved for liver cancer by the FDA		Most of the drugs were ineffective or effective only in select lines of organoids.	Li et al. (2019) [200]
Lung Adenocarcinoma	Library of 24 anti-cancer drugs		Sensitivity to each particular drug was consistent between different passages; there was a relation between some drug sensitivities and mutational profiles.	Li et al. (2020) [201]
PDO colon cancer	Curcumin	Presto Blue assay kit. Flow cytometry	Curcumin suppressed cell viability of CRC organoids in a concentration-dependent manner.	Elbadawy et al. (2021) [202]
Glioblastomas	Personalized therapies CAR-T cell immunotherapy	PDO were analyzed for invasion and proliferation of T cells, tumor cell death, and EGFRvIII antigen loss	CAR-T cells are fairly specific to their target and were unable to completely eradicate all tumor cells under our conditions.	Jacob et al. (2020) [203]
Neck squamous cell carcinoma	Panel of drugs including cisplatin, carboplatin, cetuximab, and radiotherapy in vitro	Determination of IC50	Organoid showed a higher sensitivity to carboplatin than to cisplatin; cetuximab did not act as a radiosensitizer in PDO.	Drieshuis et al. (2019) [204]
Breast cancer organoids	Small set of drugs targeting the HER signaling pathway.	cell viability and identify IC50	In vitro drug screens were consistent with in vivo xeno-transplantations and patient response.	Sachs et al. (2018) [88]
PDO prostate cancer	Enzalutamide, everolimus, and BKM-120	Luminescent Cell Viability Assay, determination of IC-50	Organoid lines suggest that drugs targeting of suppressor pathway p53 should become a therapeutic priority.	Gao et al. (2014) [89]

## Data Availability

The data presented in this study are available on request from the corresponding author.

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
