# Peer review of "Advances in Tumor Organoids for the Evaluation of Drugs: A Bibliographic Review"

_pharmaceutics, 2022, doi:10.3390/pharmaceutics14122709_

Round 1

Reviewer 1 Report

This review seems interesting, but some concerns remain. Some sentences or descriptions should be added for the revision.  Therefore, major revisions should be made before re-submission. The paper would be re-considered only when all the comments were responded to and reflected in a revised version.

The reviewer’ comments are below.

1.

What is the purpose of this review? I think it is new.

2.

Table 1

More many papers have been reported. The table is not indicated the overall research. The authors should add the research utilizing materials. The property or characteristics of each material should be described. In addition, other representative materials are noted, such as chitosan, gelatin, PLA, or PGA. In the current version, the review cannot be accepted. The table is not comprehensive.

 I recommend the paper to be quoted.

To describe each material (review)

Review

Cancers 202012(10), 2754

Tissue Eng. Part B Rev. 201016, 351–359.

To add research papers

Chitosan

https://doi.org/10.1016/j.actbio.2016.06.014

Gelatin
Tissue Eng. Part C Methods
 201925, 711–720. https://doi.org/10.1089/ten.tec.2019.0189

Silk

ACS Appl. Mater. Interfaces 2018, 10, 43, 36641–36651

3.

What is MEC?

The abbreviation should be clarified.

4. Figure 3

What is the message of this figure? This is well-known and not novel. The authors should describe the function or relationship with each cell in this figure.

Author Response

Comment

Response

1.What is the purpose of this review? I think it is new.

The aim of the review links the description of the techniques and materials development of tumor organoids, emphasizing the physicochemical properties of materials to biomimicry tumor extracellular matrix, and an extensive bibliographic review to identify the tools used for the characterization and evaluation of correspondence with the systems they represent. Moreover, this paper relates the advantages of organoids compared to other models for drug evaluation are also presented, literature that we agree is new and is not reported as is presented here. A proper correction was done to the Abstract.

2. (Table 1) More many papers have been reported. The table is not indicated the overall research. The authors should add the research utilizing materials. The property or characteristics of each material should be described. In addition, other representative materials are noted, such as chitosan, gelatin, PLA, or PGA. In the current version, the review cannot be accepted. The table is not comprehensive.

Table 1 was supplemented and it specifies the main characteristics of each material.

 I recommend the paper to be quoted.

To describe each material (review): Review: Cancers 2020, 12(10), 2754, Tissue Eng. Part B Rev. 2010, 16, 351–359. To add research papers Chitosan:https://doi.org/10.1016/j.actbio.2016.06.014,
Gelatin: Tissue Eng. Part C Methods 2019, 25, 711–720. https://doi.org/10.1089/ten.tec.2019.0189, Silk ACS Appl. Mater. Interfaces 2018, 10, 43, 36641–36651.

The paper was added to the references, thanks for the suggestion.

3.What is MEC?The abbreviation should be clarified.

It was a misspelling problem, it is ECM. The abbreviation was corrected

 4. (Figure 3) What is the message of this figure? This is well-known and not novel. The authors should describe the function or relationship with each cell in this figure.

The description of each cell was added to the main document at the end of the section.

Reviewer 2 Report

The authors present a good collection of data concerning the  development of the organoids research area.

I have some a few comments:

The abstract should be reformulated for a better clarity.

It should be helpful to make a larger description spheroids versus organoids (and tumoroids) since  the distinction is quite difficult to be done between these  systems. A figure would help too.

Table 3 contains only a few examples of anticancer drugs tested in different types of patient-derived organoids. As the title announces, I would have expected many more examples.

Author Response

Comment

Response

The abstract should be reformulated for a better clarity.

The abstract was corrected

It should be helpful to make a larger description spheroids versus organoids (and tumoroids) since the distinction is quite difficult to be done between these systems. A figure would help too.

The figure was done and included in the main text

Table 3 contains only a few examples of anticancer drugs tested in different types of patient-derived organoids. As the title announces, I would have expected many more examples.

The table was complemented.

Round 2

Reviewer 1 Report

The manuscript could be accepted.